# Agronomic Performance in Low Phytic Acid Field Peas

**DOI:** 10.3390/plants10081589

**Published:** 2021-08-02

**Authors:** Donna L. Lindsay, Ambuj B. Jha, Gene Arganosa, Raymond Glahn, Thomas D. Warkentin

**Affiliations:** 1Crop Development Centre, Department of Plant Sciences, University of Saskatchewan, Saskatoon, SK S7N 5A8, Canada; donna.lindsay@usask.ca (D.L.L.); ambuj.jha@usask.ca (A.B.J.); gene.arganosa@usask.ca (G.A.); 2Robert W Holley Ctr Ag & Health, USDA-ARS, Cornell University, Ithaca, NY 14853-2901, USA; Raymond.Glahn@ARS.USDA.GOV

**Keywords:** field pea, low phytic acid, iron bioavailability

## Abstract

Field pea is a pulse that delivers high protein content, slowly digestible starch and fiber, and many vitamins and minerals, including iron. Naturally occurring plant phytic acid molecules bind iron, lowering its availability for absorption during digestion. Two low phytic acid (*lpa*) pea lines, 1-2347-144 and 1-150-81, developed by our group had 15% lower yield and 6% lower seed weight relative to their progenitor cultivar. Subsequently, we crossed the two *lpa* lines and two cultivars, and derived 19 promising *lpa* pea breeding lines; here we document their agronomic performance based on 10 replicated field trials in Saskatchewan. Seventeen of these *lpa* lines yielded greater than 95% of the check mean (associated cultivars) and 16 were above 98% of the check mean for 1000 seed weight. The 19 *lpa* lines showed 27 to 55% lower phytic acid concentration than the check mean. Iron concentrations were similar in all the *lpa* lines and cultivars, yet the Caco-2 human cell culture assay revealed 14 of the 19 *lpa* lines had 11 to 55% greater iron bioavailability than check means. Thus, a single round of plant breeding has allowed for closing the gap in performance of low phytic acid pea.

## 1. Introduction

As a pulse, field pea (*Pisum sativum* L.) has the potential to be a high protein, low-input contributor to meeting the increasing food demands of a growing global population. In 2020, the Canadian field pea harvest was estimated at 4.6 MT, approximately 24.5% higher than the 10-year average of 3.7 MT and Saskatchewan led provincial harvest with 2.5 MT [1]. In addition to meeting yield demands, there is an ongoing desire to develop agricultural products that address nutritional deficiencies [2,3] and field pea may be part of the solution.

An essential first step to increasing macro and micronutrients in our food is to care for our soils and be sure nutrients are present for plants to incorporate in their tissues [4]. Commercial micronutrient fertilizers utilize synthetic chelators or aminochelates to protect minerals, such as iron, from bonding to oxygen or hydroxide, with the minerals released within the plant tissues [5]. While increasing the content of micronutrients in plants, referred to as biofortification, is a common approach for improving nutritional quality of food [6], another strategy is limiting the action of molecules that can negatively impact nutrient bioavailability. Phytic acid, a naturally occurring plant molecule, has been recognized in pulses, including peas, since 1903 [7]. The phosphorus-based molecule phytic acid strongly chelates mineral elements into the insoluble salt phytate, rendering the nutrients inaccessible during digestion by humans and other monogastric animals. Raboy [8] provides a holistic overview of the positive consequences of breeding low phytic acid (*lpa*) crops, including: health benefits from increased bioavailability of phosphorus, potassium, calcium, zinc, and iron in digestion by humans, poultry, swine, and fish; sustainable phosphorus management that reduces fertilizer application for crops; and reduced eutrophication, i.e., lessening environmental pollution from undigested phosphorus. While micronutrient availability in *lpa* crops has the potential to provide health benefits, especially to pregnant women and young children [8], it should also be noted that lowering phytic acid levels may reduce the beneficial effects this molecule has shown in prevention of cardiovascular disease and kidney stone formation, inhibition of cancer, and protection from effects of inflammatory bowel disease [9].

When genetic pathways capable of increasing bioavailability of micronutrients exist in a crop of interest, conventional plant breeding is a sustainable approach to improving the health status of people globally [10,11]. Our lab has established two *lpa* pea lines: 1-2347-144 and 1-150-81. An important aspect of genetic research in *lpa* lines is the identification of specific genes and pathways affected. Phytic acid biosynthesis is catalyzed by D-myo-inositol 3-phosphate synthase (MIPS) and the gene encoding this enzyme has been identified as a common target in *lpa* grains [12]. Utilizing Mendelian-style analysis, our lab established that the mutation in our two *lpa* lines is recessive and probably allelic [13]; however, a mutation in *MIPS* was not present [14]. Therefore, further investigation is warranted to identify the target gene in these *lpa* pea mutants in support of breeding *lpa* lines in other crops. 

A focus on field pea nutrition, with their high protein content, slowly digested carbohydrates, dietary fibre, and concentration of vitamins and minerals, including iron, can be explored within the broader plant breeding vision of maintaining yield. The objective of this paper is to update the status of our ongoing breeding initiative to develop pea cultivars with lower phytic acid concentrations that maintain strong agronomic characteristics.

## 2. Results

### 2.1. Low Phytic Acid (lpa) Pea Breeding Lines: Agronomic and Seed Quality Performance

Analysis of variance was conducted on the combined data of 24 pea lines (19 low phytic acid (*lpa*) breeding lines, 4 check cultivars (CDC Bronco, CDC Raezer, CDC Limerick, and CDC Amarillo), and one *lpa* mutant (1-2347-144)). These lines were evaluated at 10 station years from 2017 to 2020 at Kamsack, Meath Park, Rosthern, and Saskatoon locations in Saskatchewan, Canada. Yield was measured for all 10 station years; other characteristics were observed for two to nine of the station years (environment). The effects of line and environment were significant (*p* < 0.05) for all the reported traits, with the exception of the effect of line on Zn concentration (Table 1). Significant differences due to the line X environment interaction were observed for four of 11 traits. For protein, phytic acid, and iron bioavailability, lines contributed a greater proportion of the phenotypic variance than environment, whereas for the other traits, the contribution of environment was greater compared to that of the lines. 

For the 24 pea lines, a wide range of variation was observed for phytic acid concentration (0.5 to 2.5 mg/g sample), iron bioavailability (33 to 99 ng ferritin/mg protein), protein concentration (22 to 31% of the seed dry matter), grain yield (2,052 to 7,222 kg/ha), and other traits (Table 1). For comparisons, a check mean was determined for CDC Bronco, CDC Raezer, and CDC Limerick, as these cultivars contributed genetic backgrounds to the breeding lines. In general, *lpa* breeding lines were taller (18%) than the check mean, whereas stand density, lodging, maturity, and grain yield, as well as protein, iron, zinc, and phosphorus concentration were similar compared to the check cultivars (Table 1). Phytic acid concentration was 30 to 55% lower for the *lpa* breeding lines than the check mean. Seven *lpa* breeding lines showed 25% to 55% higher iron bioavailability, as measured by the Caco-2 bioassay compared to the check mean.

Analysis of variance for station years indicated as ‘environment’ (x 2 reps each). Data in table is means. Minimum and maximum are for raw data. Agronomic data: stand density at flowering 1–9, where 1 is very poor plant stand density and 9 is full plant stand density (St Den); height as average of two random plants at maturity (HT (cm)); Lodging at maturity 1–9, where 1 is upright and 9 is completely horizontal (Lodg); plot maturity 1–3 taken at early maturity, where 1 is early and 3 is late (Mat); Yield (kg/ha); yield as % check mean (CDC Bronco, CDC Raezer, CDC Limerick); 1000 seed weight (SDWT (g/1000)); near-infrared reflectance assessment of protein (%); element concentrations by ICP-ES and ICP-MS for phosphorus (P), zinc (Zn), and iron (Fe) (µg/g); mg phytic acid-phosphorus per g sample, as measured by Wade’s assay (mg PA-P/g sample); iron bioavailability as measured by Caco-2 bioassay (ng ferritin/mg protein).

### 2.2. Low Phytic Acid (lpa) Pea Breeding Lines: Correlation Analysis of Agronomic and Seed Quality Traits

Pearson correlation analysis indicated phytic acid-phosphorus concentration was negatively correlated with iron bioavailability (r = −0.45, *p* < 0.05) (Table 2). Iron bioavailability was negatively correlated with grain yield (r = −0.49, *p* < 0.05). Fe concentration was positively correlated with Zn concentration (r = 0.66, *p* < 0.001), whereas Fe and Zn concentrations were positively correlated with protein (*p* < 0.001) and phytic acid-phosphorus concentration (*p* < 0.001). Phytic acid-phosphorus concentration was positively correlated with protein concentration (r = 0.42, *p* < 0.05) and negatively correlated with grain yield (r = −0.42, *p* < 0.05). Stand density was positively correlated with maturity, yield, and protein, as well as Zn and Fe concentration. Stand density was negatively correlated with lodging.

### 2.3. Evaluation of Low Phytic Acid (lpa) Pea Breeding Lines Arising from Independent Genetic Origins

T-test was conducted to compare the differences in phytic acid-phosphorus concentrations and iron bioavailability in the two genetic backgrounds, i.e., 4802 (7 breeding lines) and 4803 (12 breeding lines). Results indicated significant differences for protein (*p* < 0.01), Zn concentration (*p* < 0.01), phytic acid-phosphorus concentration (*p* < 0.001), and iron bioavailability (*p* < 0.05) between the 4802 and 4803 breeding lines (Table 3).

## 3. Discussion

Plants utilize phytic acid as an important reservoir of phosphorus, a mineral essential to healthy embryo development, seed quality, and high yield. Phytic acid also chelates cations, such as iron and zinc, as the insoluble salt phytate. Human uptake of micronutrients such as iron and zinc benefits from lower phytic acid concentrations in grains and pulses. A crop breeding challenge is to reach and maintain a balance between sufficient, but not excess, phytic acid concentrations for mutual healthy plant development and increased bioavailability of micronutrients for human and monogastric livestock health [2,8]. 

In their review paper of low phytic acid (*lpa*) mutants in crops, Colombo et al. [15] noted significant negative pleiotropic effects on field performance. For example, *lpa* mutants in wheat, maize, barley, and soybean have shown one or more of the following: suppressed germination, seed weight reduction, and susceptibility to abiotic and biotic stresses [16,17,18,19]. In contrast to these reports, many of the *lpa* pea breeding lines of this study showed stand density, height, seed weight, and yield at the same or higher levels than cultivar parents. While we did not directly test for stress responses, the high yield in several field environments suggests the lines were not susceptible. An earlier exception to the negative effects of *lpa* in crops was observed in the *lpa*1 mutant in common bean, which was found to have equal or superior agronomic performance compared to wild type [20,21]. Our study suggests another pulse, field pea, can be bred to sustain lower phytic acid concentrations without negative impact on important agronomic traits such as stand density, time to maturity and lodging. Yield, as a reflection of seedling emergence, plant development, disease resistance, and successful maturation, did not significantly differ in most of the *lpa* pea breeding lines of this study relative to commercial cultivars. 

Having determined crop health, an important question in this study was whether the phytic acid levels in the *lpa* pea breeding lines were sufficiently low to affect iron bioavailability. The Caco-2 cell culture assay which was designed to study the intestinal absorption of iron had previously shown 1.4 to 1.9 higher iron bioavailability in our *lpa* pea mutant lines 1-2347-144 and 1-150-81 [22]. Here, we again utilized the Caco-2 bioassay to now test 19 *lpa* pea breeding lines developed from earlier crosses between the mutants and two cultivars [23]. Replicated samples from field trials of 2018 showed significantly greater iron absorption for 17 of the 19 *lpa* breeding lines than the relevant checks. 

It is also noteworthy to consider the iron bioavailability of peas relative to other pulses, as measured by the Caco-2 cell bioassay. For example, recent measurement of iron bioavailability in yellow beans yielded Caco-2 cell ferritin formation values of approximately 15 ng ferritin/mg cell protein [24]. In another study of common beans, Caco-2 cell ferritin values of slow darkening pinto beans averaged 9–13 ng ferritin/mg cell protein [25]. Iron concentration of the beans in the above studies was approximately 66–75 µg/g. The bean varieties of these studies are considered to be relatively high in iron bioavailability for beans. For lentils, Caco-2 cell ferritin values are similar to beans but can be higher depending on variety, location, and processing [26]. By comparison, Caco-2 cell ferritin formation values from the pea varieties in this paper ranged from 53–87 ng ferritin/mg cell protein, and iron concentration of the peas was significantly less at 28–36 µg/g. Thus, the results clearly indicate that peas can deliver substantially more bioavailable iron per gram of food relative to other pulses.

The nutritional value of the *lpa* pea breeding lines was further evidenced in a feeding study utilizing chickens (*Gallus gallus*) [27]. Collaborating with Dr. Elad Tako, Cornell University, Ithaca, NY, chickens were fed four of our *lpa* pea breeding lines or two control diets. Several characteristics were monitored, including body hemoglobin iron, body weight, and energetic status. Upon completion of the study, hepatic iron and ferritin, pectoral glycogen, duodenal gene expression, and cecum bacterial population analyses were conducted. The data indicated that *lpa* pea lines could moderately improve iron status. There were also significant effects on duodenal brush border membrane functionality and intestinal microbiota. Specifically, expression of the duodenal brush border membrane gene ferroportin (*FPN*) was significantly upregulated and there was an increase in abundance of beneficial gut bacteria in chicks fed *lpa* peas [27].

Having documented higher iron bioavailability in *lpa* pea breeding lines via the Caco-2 bioassay and a chicken feed study [27], the next logical step is human trials. To that end, we are currently running a pilot study feeding *lpa* peas to athletic young women who are low in iron. 

## 4. Materials and Methods

### 4.1. Low Phytic Acid (lpa) Breeding Lines

The pea breeding program at the Crop Development Centre, University of Saskatchewan developed breeding lines with a low phytic acid (*lpa*) trait relative to cultivar controls. Two *lpa* pea lines (1-2347-144 and 1-150-81) were generated by non-destructive screening with Chen’s reagent following chemical mutagenization of the cultivar CDC Bronco [28,29]. To increase agronomic performance and to expand the yellow cotyledon background to include the green cotyledon market class, the two *lpa* lines were crossed with cultivars in 2011 [23], i.e., cross 4802 = 1-2347-144 × CDC Raezer [23,30] and cross 4803 = 1-150-81 × CDC Limerick [23,31]. Progeny with lower phytic acid concentrations than the cultivar parents were selected in the F_2_ generation. Promising candidates were advanced to F_5_, then 400 lines from each cross were evaluated for phytic acid concentration by non-destructive seed testing. In 2016, ten F_6_ lines from each of four categories (low phytic acid and normal phytic acid x two cotyledon colours [23]) were tested for iron bioavailability via Caco-2 bioassay and 19 F_6_ breeding lines were chosen to take forward in the field trials of this study. 

The chosen 19 breeding lines were evaluated in the field for F_6_–F_9_ generations, plus CDC Bronco, 1-2347-144, CDC Raezer, CDC Limerick, and the popular cultivar CDC Amarillo. This 24 line group was tested at multiple locations over multiple years for a total of 10 station years (2017 to 2020) to determine the effects of genotype, environment, and the genotype x environment interactions. Nutritional traits were evaluated by measuring protein, iron, and phytic acid concentration, and the Caco-2 human cell culture bioassay again assessed iron bioavailability (2018 samples).

### 4.2. Field Trials

From 2017 to 2020, in agricultural fields near the communities of Meath Park, Rosthern, Kamsack, and Saskatoon in Saskatchewan, Canada (for a total of 10 station years), two replicates of the 24 lines were seeded in a randomized complete block design of 3.6 m × 1 m plots. Spring seeding was conducted at a rate of 90 seeds m^2^. Plots were supplemented with solid core granular inoculant *Rhizobium leguminosarum biovar viceae* at manufacturer’s recommended rate of 3.0 kg/ha (Nodulator^®^ XL SCG, San Carlos, CA, USA). Weeds were controlled with registered herbicide applications tailored to the site, supplemented with hand weeding. The plots were mechanically harvested in August when the majority of the plants were physiologically mature and dried by chemical desiccation.

For the 10 station years, grain yield was measured and calculated for each plot and then converted to kg/ha. Other agronomic and nutrition data collected for station years varied but could include: stand density, plant height, lodging, maturity, 1000 seed weight, protein prediction by near infrared (NIR) analysis [32], and elemental concentration of phosphorus, zinc, and iron by ICP-ES or ICP-MS [33]. Phytic acid concentrations were calculated utilizing Wade’s assay [34] and iron bioavailability was determined by the Caco-2 bioassay [35]. Data collected for some station years but not reported here included days to flowering, days to maturity, flower colour, leaf type, NIR prediction of starch, fibre, oil, and ash, seed coat breakage, and seed phenotypes of shape, dimpling, bleaching, and greenness.

### 4.3. Assessment of Phytic Acid-Phosphorus by Wade’s Assay

Modified Wade’s reagent method [34] was used to assess phytic acid-phosphorus levels in the 24 lines of this study. Briefly, 1.0 mL of 0.8 N HCl was added to 0.05 g (<0.5 mm sieve) of pea flour in individual microcentrifuge tubes then shaken with a Labquake Shaker (Thermo Scientific), overnight at room temperature. Samples were centrifuged at 8000 rpm for 20 min. Ten μL of supernatant were transferred to a new microcentrifuge tube containing 740 μL of double distilled H_2_O and 250 μL of modified Wade’s reagent (0.3% sulfosalicylic acid + 0.03% FeCl_3_·6H_2_O). A 1000 µg g^2^ phytic acid stock (549.9 mg of phytic acid sodium salt hydrate in 100 mL of 0.8 N HCl) was diluted into 25, 50, 100, 200, 300, 400, 500 and 600 µg g ^2^as calibration standards. The samples and standards were vortexed, then 200 μL of each solution was transferred into a 96 well flat bottomed tissue culture plate and read at 490 nm using a BioRad xMark microplate spectrophotometer (Bio-Rad Laboratories, Ltd., Mississauga, ON, Canada). Phytic acid-phosphorus (mg PA-P/g sample) concentration in seed was calculated from the absorbance reading of spectrophotometer divided by 50.

### 4.4. Assessment of Iron Bioavailability by Caco-2 Cell Bioassay

Twenty grams of raw pea seed representing 96 samples (24 lines × 2 plot reps × 2 locations) grown in 2018 were placed in 15 cm diameter foil pie plates, submerged in 60 mL 18 Megaohm water, and soaked overnight at room temperature. Peas that did not imbibe were removed. To cook the peas, aluminum foil was secured as lids and the plates were autoclaved on liquid cycle at 121 °C for 30 min. The samples were cooled to room temperature, moved to a −80 °C freezer, and then freeze-dried following manufacturer instructions (Labconco ™, Kansas City, MO, USA). The samples were ground to powder with a steel blade coffee grinder, being careful not to heat the samples. 

The Caco-2 cell culture bioassay test was conducted in the laboratory of Dr. Raymond Glahn, USDA-ARS, Ithaca, NY [35,36]. For digestion, 0.5 g of cooked ground seed was added to 10 mL solution of 140 mM NaCl and 5 mM KCl. After mixing well, the solution was adjusted to pH 2 with 0.1 M HCl, then 0.5 mL pepsin solution was added. Rocking incubation was performed for 1 h. The solution was then adjusted to pH 5.5 to 6.0 with 1.0 M NaHCO_3_. 2.5 mL of pancreatic-bile solution was added, followed by adjusting pH to 6.9 to 7.0 with 1.0 M NaHCO_3_. Both pepsin solution and pancreatic-bile solution were purified by cation exchange resin (Chelex 100, Bio-Rad Laboratories, Inc., Hercules, CA, USA). After digestion, the Caco-2 cells were fed 1.5 mL digested sample through a 15 kDa cutoff dialysis membrane. The cells were incubated and rocked gently for 2 h. The digested sample solution and membrane were removed and the cells were placed back in the incubator without rocking for 22 h. Finally, cells were harvested for analysis of total protein using a colorimetric assay (DC Protein Assay, Bio-Rad Laboratories, Inc., Hercules, CA, USA) and for analysis of ferritin using an immunoradiometric assay (Fer-Iron II, Ramco Laboratories, Inc., Stafford, TX, USA) by manufacturers’ instructions. Iron bioavailability (ng ferritin/mg protein) for samples from different experimental runs was standardized by a standard lentil sample included in each run.

### 4.5. Statistical Analysis

Analysis of variance (ANOVA) was conducted using PROC MIXED in SAS 9.4 (SAS Institute Inc., Cary, NC, USA) using data from 2 to 10 station years (2017–2020). For this analysis, each station-year was considered an ‘environment’. The 19 breeding lines were considered as a fixed effect and replication as a random effect. Pearson correlation between traits was determined using PROC CORR implemented in SAS. PROC TTEST was used to evaluate differences between the two genetic backgrounds, 4802 (7 breeding lines) and 4803 (12 breeding lines).

## 5. Conclusions

Encouraging consumption of plant-based meals is an efficient and environmentally positive way to feed a growing global human population [37,38]. Increased consumption of peas could be an important factor in that scenario. An objective of this research, as part of the legacy of the Crop Development Centre (CDC) (established in 1971), University of Saskatchewan is to promote health and well-being by developing and delivering nutritious crops. A single round of breeding closed the yield gap between the previously developed *lpa* pea mutants and relevant check cultivars. Current *lpa* pea breeding lines maintained protein concentrations and increased bioavailability of iron (and, although not measured here, almost certainly other nutrients bound to phytic acid, such as zinc and calcium), without sacrificing yield or other important agronomic traits. The *lpa* trait has been incorporated into pea cultivars that should be ready for release to growers in the near future. 

## Figures and Tables

**Table 1 plants-10-01589-t001:** Analysis of variance and mean values for agronomic and seed quality traits of 24 pea varieties evaluated over 2 to 10 station years in Saskatchewan, Canada, between 2017–2020.

Line Name	St Den	HT	Lodg	Mat	Yield	%Check	SDWT	Protein	P	Zn	Fe	mg PA-P /g Sample	ng Ferritin /mg Protein
	1 to 9	cm	1 to 9	1 to 3	kg/ha	Mean	g/1000	%	µg/g	µg/g	µg/g		
CDC Amarillo	8.4	74	3.1	1.8	4181	111	225	24.3	3428	27.9	45.3	1.29	67.74
CDC Raezer	8.3	70	4.1	1.4	3522	93	206	24.7	4174	31.3	49.8	1.95	54.60
CDC Limerick	8.0	70	3.9	1.8	3748	99	198	27.0	3828	32.8	49.7	1.65	61.02
CDC Bronco	8.3	68	4.4	1.8	4080	108	213	25.6	3684	31.9	49.1	1.48	53.87
1-2347-144	8.3	61	4.3	1.8	3548	94	196	25.8	3678	30.0	49.2	0.85	61.96
4802-8-60G-L	8.3	71	3.2	2.0	3356	89	232	25.5	3967	30.3	47.1	0.96	75.70
4802-8-4G-L	8.4	76	3.3	2.5	3872	102	202	26.7	4008	29.7	51.7	0.76	61.77
4802-8-163G-L	8.7	81	3.4	2.5	3854	102	177	25.8	3768	30.8	50.3	0.88	67.65
4802-8-46Y-L	8.2	72	4.0	1.1	3292	87	225	25.2	3801	30.5	50.9	1.02	87.43
4802-8-87Y-L	8.3	83	4.2	1.9	3621	96	211	26.0	3819	32.7	47.6	0.86	72.90
4802-8-1Y-L	8.4	70	4.4	1.6	3766	100	195	27.0	3985	31.2	52.4	0.92	81.03
4802-8-85Y-L	8.4	80	3.9	2.4	3872	102	192	26.1	3882	31.9	50.6	0.93	74.45
4803-4-78G-L	8.6	78	3.6	2.1	3657	97	224	28.3	3936	32.8	51.1	1.14	67.61
4803-4-29G-L	8.8	76	3.4	2.1	3882	103	216	28.1	4141	35.2	54.8	1.13	70.76
4803-4-70G-L	8.8	75	3.2	2.0	3846	102	216	27.5	4046	36.4	52.7	1.06	71.27
4803-4-26G-L	8.2	67	3.9	2.3	4008	106	207	25.6	3914	31.5	46.8	1.24	66.99
4803-4-74G-L	8.8	76	3.2	1.9	3686	97	226	27.9	4228	34.1	52.8	1.07	69.94
4803-4-30G-L	8.6	74	3.3	2.1	3763	99	225	28.1	4087	33.2	51.4	1.06	66.59
4803-4-31G-L	8.7	73	3.6	2.1	3857	102	204	26.1	3980	34.3	50.7	1.00	65.07
4803-4-9Y-L	8.4	67	3.9	1.8	3865	102	221	27.5	3653	31.3	50.2	1.08	59.08
4803-4-43Y-L	8.6	71	3.7	2.4	4257	113	207	25.9	3479	29.4	48.1	1.10	53.26
4803-4-27Y-L	8.4	72	3.6	1.9	3834	101	220	27.4	3665	31.2	48.4	1.12	57.08
4803-4-92Y-L	8.2	69	4.1	1.9	3868	102	217	27.2	3963	33.6	51.5	1.02	59.24
4803-4-71Y-L	8.8	76	3.3	2.1	3745	99	210	27.3	3819	32.9	51.2	1.11	62.83
**Environment**	8	3	9	8	10		4	7	5	5	4	5	2
**Variety**	*** (12%)	* (10%)	*** (9%)	*** (18%)	*** (5%)		*** (10%)	*** (42%)	*** (6%)	ns (2%)	*** (8%)	*** (29%)	*** (28%)
**Environment**	*** (9%)	*** (38%)	*** (43%)	*** (41%)	*** (78%)		*** (47%)	*** (26%)	*** (52%)	*** (81%)	*** (43%)	*** (8%)	*** (16%)
**Variety x Environment**	** (14%)	ns (0%)	** (9%)	** (9%)	* (2%)		ns(0%)	**(6%)	ns(0%)	ns(0%)	ns(0%)	ns(0%)	ns(2%)
**Minimum**	7.0	38	2.0	1.0	2052		133	22.1	2642.3	15.4	39.8	0.5	32.8
**Maximum**	9.0	95	7.0	3.0	7222		267	31.4	5094.2	55.5	66.3	2.5	99.2
**Overall Mean**	8.4	73	3.7	2.0	3792		211	26.5	3791.8	32.3	49.7	1.11	66.24
**SD**	0.5	11.5	1.1	0.7	890.6		33.1	1.6	607.2	10.0	5.5	0.4	12.4
**CV**	6,4	15.8	30.5	36.6	23.5		15.7	6.2	16.0	30.9	11.0	37.6	18.7
**LSD_0.05_**	0.31	11.0	0.47	0.29	226.1		25.7	0.63	212.1	6.0	2.8	0.34	13.51

ns, not significant; *, *p* ≤ 0.05; **, *p* ≤ 0.01; ***, *p* ≤ 0.001; percentage in parentheses indicates phenotypic variance; SD, standard deviation; CV, coefficient of variation; LSD, least significant difference.

**Table 2 plants-10-01589-t002:** Pearson correlation analysis of performance traits of 24 pea lines evaluated from 2017 to 2020 at Kamsack, Meath Park, Rosthern, and Saskatoon locations in Saskatchewan.

Characteristic	Stand Density	Lodging	Maturity	Height	Yield	SDWT	Protein	P	Zn	Fe	Phytic Acid-P
**Lodging**	−0.69 ***										
**Maturity**	0.43 *	−0.41 *									
**Height**	0.55 **	−0.48 *	0.42 *								
**Yield**	0.29 ns	−0.13 ns	0.52 **	0.00 ns							
**SDWT**	0.04 ns	−0.36 ns	−0.35 ns	−0.06 ns	−0.19 ns						
**Protein**	0.52 **	−0.23 ns	0.26 ns	0.20 ns	0.00 ns	0.21 ns					
**P**	0.24 ns	−0.16 ns	0.00 ns	0.22 ns	−0.43 *	0.09 ns	0.42 *				
**Zn**	0.48 *	−0.14 ns	0.09 ns	0.27 ns	−0.09 ns	0.13 ns	0.65 ***	0.65 ***			
**Fe**	0.51 *	−0.09 ns	0.05 ns	0.21 ns	−0.17 ns	−0.08 ns	0.69 ***	0.66 ***	0.66 ***		
**Phytic acid-P**	−0.14 ns	0.16 ns	−0.40 *	−0.27 ns	0.08 ns	0.09 ns	−0.24 ns	0.05 ns	0.03 ns	−0.18 ns	
**Iron bioavailability**	0.02 ns	−0.08 ns	−0.23 ns	0.35 ns	−0.49 *	0.11 ns	−0.01 ns	0.27 ns	0.09 ns	0.21 ns	−0.45 *

ns, not significant; *, *p* ≤ 0.05; **, *p* ≤ 0.01; ***, *p* ≤ 0.001; SDWT, 1000 seed weight.

**Table 3 plants-10-01589-t003:** T-test comparison of the 4802 and 4803 genetic backgrounds in the 19 *lpa* breeding lines.

Cross Number	mgPA-P/g Sample	ng Ferritin/mg Protein
**4802**	0.9	74.4
**4803**	1.1	64.1
	***	*

* *p* < 0.05; *** *p* < 0.001. 4802 are the 7 breeding lines arising from the cross: CDC Raezer x *lpa* 1-2347-144; 4803 are the 12 breeding lines arising from the cross: CDC Limerick x *lpa* 1-150-81.

## Data Availability

Not applicable.

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
