# Peer review of "Agronomic Performance in Low Phytic Acid Field Peas"

_plants, 2021, doi:10.3390/plants10081589_

Round 1
Reviewer 1 Report
The manuscript describes the agronomic characterization of 19 low phytic acid field pea breeding lines. 17 of them show good agronomic performance. Seed iron content is similar to control lines while its bioavailability is increased as expected due to reduction in phytic acid content.
The data are interesting and the manuscript is well written. However, in my opinion data concerning percentage of seed germination of the different lines is required to complete the characterization.
Minor suggestions:
lpa always italics when eeferring to mutation, mutant lines;
page 5, penultimate sentence: please add "seeds" after "400"
Author Response
We appreciate the helpful suggestions of the two reviewers. Our detailed replies follow preceded in each case by TW.
Reviewer One
The manuscript describes the agronomic characterization of 19 low phytic acid field pea breeding lines. 17 of them show good agronomic performance. Seed iron content is similar to control lines while its bioavailability is increased as expected due to reduction in phytic acid content.
The data are interesting and the manuscript is well written. However, in my opinion data concerning percentage of seed germination of the different lines is required to complete the characterization.
TW: We addressed the germination rate by adding plant stand density data. These data are regularly collected on all field plots.
Minor suggestions:
lpa always italics when referring to mutation, mutant lines;
TW: We thought we had consistently italicized the mutant name, but apparently some formatting was lost and we appreciate you pointing this out.
page 5, penultimate sentence: please add seeds after 400
TW: We chose to add “lines” after the 400 as it more accurately describes what we developed.
Reviewer 2 Report
Respect the suggestions. Recommendations, suggestions and comments are given at the edge (side) of the article.

Author Response
We appreciate the helpful suggestions of the two reviewers. Our detailed replies follow preceded in each case by TW.
Reviewer Two
I believe the combination of the words pulse and crops is not commonly used in the European environment
TW: Thank you – we dropped ‘crop’ from the three places we used the phrase pulse crop
Fill the introduction chapter with information about the positive effect of phytic acid (kidney stones, cancer)
TW: We added a reference to the positive health aspects of phytic acid: review by Silva and Bracarense 2016.
So far, I have not encountered the titles of the subchapter results being named according to statistical method of evaluation. These subchapters are usually named according to what is being studied.
TW: We addressed this by modifying subchapter headings.
Below the table, provide explanations of all abbreviations used (HT, Lodg? Mat, etc) although they are explained below in chapter Materials and Methods
TW: Most of the abbreviations were explained, but we did tidy the order and make sure is was complete.
The Wade test must be described in the Materials and Methods
TW: We describe the Wade’s test on page 3.
Chelation of nutrients in fertilizers achieves better utilization of microelements (metals) by plants from fertilizers. The same is true in animal production. Chelation of feeds = better use of Fe, but also other nutrients in animals.
TW: We added a sentence about the difference between fertilizer chelators and non-soluble phytate
In the discussion chapter, authors should focus on explaining, justifying the results they have achieved and comparing them with those already achieved by other authors. These 5 lines should go to the introduction chapter
TW: We chose to leave the subject of pleiotropic effects in lpa crops in the Discussion, with modifications, as the goal of our paper is to show that these lpa lines essentially do not suffer from these negative pleiotropic effects.
I recommend adding authors to the methods used in the experiment. Then list the authors in the references.
TW: We added the appropriate references to Methods.
A typo space between words in 4.5 Stat Analysis
TW: Thank you for catching this. We corrected the typo.
Round 2
Reviewer 1 Report
I think that the authors properly modified and improved their manuscript.